# Improved assembly DC circuit breaker based on resonant current injection

**Puyi Cui**[1,2]*, **Guoli Li**[1,3], **Qian Zhang**[1,4], **Qinglian He**[5], **Zhong Chen**[6], **Wei Yang**[6]

**1** School of Electrical Engineering and Automation, Anhui University, Hefei, China, **2** Engineering Research Center of Power Quality, Ministry of Education, Hefei, China, **3** Anhui Key Laboratory of Industrial Energy-Saving and Safety, Anhui University, Hefei, China, **4** Anhui Collaborative Innovation Center of Industrial Energy-Saving and Power Quality Control, Anhui University, Hefei, China, **5** XJ ELECTRIC Co., Ltd, Xuchang, China, **6** Electric Power Research Institute of State Grid Anhui Electric Power Co., Ltd, Hefei, China

* qinnei028246@163.com

## Abstract

DC circuit breakers (DCCBs) with high breaking capacity and low cost are necessary for quick fault clearance in DC networks. The assembly DC circuit breakers (ADCCBs) have a main breaking section (MBS) and a sub-breaking sections (SBS) for each line, which greatly reduce the cost. But in conventional operation, it bears high voltage for a long time when there is a main switch grounding process in any line fault action. To address this problem, a multiport assembly circuit breaker based on current injection (CI-MPACB) is proposed, which is able to generate a resonant current with increasing amplitude by controlling the duty cycle of Integrated Gate-Commutated Thyristors (IGCTs). Then the resonant current is injected into the SBS to generate current zero crossing and arc extinction. A complex frequency domain circuit analysis is performed on the MBS to describe the action logic as well as the commutation characteristics. In addition, the parameters of each component of the MBS are subject to multiple constraints and reasonable design to ensure the fault current could be cut off quickly and reliably. The cost of existing design is greatly reduced due to the design idea of resonant current injection device parameter selection. Finally, a PSCAD/ EMTDC simulation confirms the opening viability of CI-MPACB and the accuracy of the parameter design. The test results show that the designed CI-MPACB can cut off DC fault lines.

## 1 Introduction

An efficient method for facilitating seamless access, dependable transmission, appropriate allocation, and flexible consumption of extensive renewable energy is the utilization of DC grids with voltage source converter (VSC) [1]. It is one of the most important technologies to sustain the construction of a new power system with renewable energy sources in China and to achieve the ambitious goal of "carbon neutrality and emission peak " [2–4]. The DC grids with VSCs are "low damping" systems in comparison to the AC systems. Both the fault current and range might expand quickly at the same time [5]. To assure a continuous and reliable

**Funding:** The study was supported by "State Grid Corporation Headquarters Science and Technology Project (5500-20220110A-1-1-ZN)" The funders

had no role in study design, data collection and analysis, decision to publish, or preparation of the manuscript.

**Competing interests:** The authors have declared that no competing interests exist.

operation, it is critical to provide fast fault current blocking of the inverters in DC network after a short circuit fault in line.

DC circuit breakers (DCCBs) are one of the most reliable, rapid, low loss and low cost solutions to achieve fault isolation in the DC grid [6–8]. As there is no natural current in DC system, DCCB isolates fast transient and high short-circuit current from HVDC converter, making DC fault interruption more difficult [9, 10]. Multiple topologies and concepts have thus far been put forth to safeguard multi-terminal DC grids [11–13]. There are three main categories of DCCBs: mechanical DCCBs (MDCCBs), solid-state DCCBs (SSDCCBs) [14–16] and hybrid DCCBs (HDCCBs) [17–21]. Mechanical DCCBs have the advantage of lower losses and cost than SSDCCBs. However, spring actuators have long operating time. SSDCCBs have a high cost due to its large use of power electronic (PE) devices. HDCCBs provide a trade-off between the fast fault interruption and low conduction losses. But a typical HDCCB contains hundreds of PE switches. A mesh HVDC grid consists of numerous converters and extensive transmission lines. As a result, there are much more HDCCBs needed, which raises the cost.

The assembly DC circuit breakers (ADCCBs) consist of a main breaking section (MBS) on each DC bus in the corresponding converter station and the sub-breaking section (SBS) of each line connected to the station. It can cut all types of faulty lines, reduce the number of MBS and the manufacturing costs [22–24]. However, this typical ADCCB has the following problems:

1. In normal operation, the MBS of the ADCCB and the auxiliary discharge switches on each line stay in the off state. At this time, each part MBS has to bear the voltage of the system to the ground. The internal PE devices can withstand higher voltage for a long time, which is harmful to the reliable of PE devices.

2. After any line fault, the MB is grounded for a long time during the action of the ADCCB.

To solve the above problems, the VSC assisted resonant current assembly DCCB (VARC-DCCB) has been proposed. Due to the shared transfer branch and energy-consuming branch between adjacent lines, the construction cost of the breakers can be significantly reduced. In reference [25], an ultra-high speed actuator and resonant circuits are used. It combines the benefits of MDCCB and HDCCBs for shorter run times, lower conduction losses and better cost effectiveness. A new multiport DCCB (MP-DCCB) based on current injection is proposed in [26]. Its fault isolation time is reduced by detouring the current limiting inductor during dissipation of energy. Accordingly, a further diode branch is built to link the transfer branch to the bus [27]. This branch is capable of eliminating bus faults or fast mechanical switching faults. The bus fault is eliminated by a bypassing branch and thyristor with diode bridge circuit loop design [28].

IGBTs or injection enhanced gate transistors (IEGTs) are frequently used to break current for high-capacity interruptions because of their significant current-breaking capacity [29, 30]. Substantial IGBT series are necessary in the above MP-DCCB transfer branch circuit as switching elements. However, the massive IGBT series give rise to a series voltage unbalance. When the fault current is larger than 15 kA, a number of IGBTs/IEGTs must be linked in parallel in order to achieve a dependable interruption with high cost. Therefore, IGCTs with high surge current withstanding capacity and low cost are seen in rise in DCCBs as an economic alternative.

Due to the existing problems of the current developed CBs, a new multiport assembly DC circuit breaker based on current injection (CI-MPACB) is proposed. Firstly, the expensive HDCCB is widely used in the project at present. It adopts the design of one common MBS and several SBSs to reduce the cost. Secondly, the traditional ADCCBs proposed at present have

the problem of the MBS needs to withstand high voltage for a long time. A novel auxiliary resonant current injection method is used. This topology solves the above problems of MBS. In addition, this paper also analyzes the topology principle of MBS and the working process of MBS and CI-MPACB in detail. Thirdly, a large number of series-parallel IGBTs are usually used as PE devices in CBs at present. This also causes high costs. The PE devices which control the current direction of the current injection branch are optimized. We propose to use fewer IGCTs combinations in place of large number of IGBT combinations. Finally, the compactness and low cost of CI-MPACB are realized to meet the requirement of fast and reliable disconnection of circuit breakers. By using multiple constraints to select the device parameters involved in the current injection branch.

The contributions of this article are manifold.

1. We propose a new type ADCCB, which is based on the design idea of common MBS. It solves the problem of high cost due to the large number of CBs which are caused by multiple converters and multiple transmission lines in reticulated HVDC system.

2. The MBS of CI-MPACB uses resonant current injection to make the line current turn off at zero crossing. It solves two problems in traditional ADCCBs: 1. PE devices bear high voltage for a long time during normal operation, 2. MB is grounded for a long time during fault removal operation. We also perform multi-constraints and cost optimization for the precharge capacitor and its voltage, resonant capacitor and resonant inductor in current injection branch. It causes compactness and low cost of CI-MPACB.

3. We use IGCTs with high surge current resistance and low cost as the conduction components of the current injection branch. The principle of this new topology is examined in detail, and the rationality of the design is finally verified according to relevant simulation. The problem of high cost caused by a large number of series-parallel IGBT combinations has been solved. And the cost comparison between the PE devices has made the effect of this optimization scheme more intuitive.

The organization of this paper is represented as following. The Topology of the DC circuit breaker section presents the topology and operating principle of CI-MPACB. The Electrical parameter design section presents the optimized design of electrical parameters of the CI-MPACB. The Scheme Comparison and Cost Analysis section compares the technicality and cost of CBs. Based on simulation results, The CASE STUDY OF CI-MPACB IN THE VSC-HVDC section verifies the effectiveness and applicability of the circuit breaker. At last, conclusions are given in The Scheme Comparison and Cost Analysis section.

## 2 Topology of the DC circuit breaker

### 2.1 Structure of the RM-DC circuit breakers

In order to realize optional fault isolation, traditional multiport DCCB shall be installed at both ends of each lines connected to the DC bus. This will lead to high construction costs. As shown in Fig 1, the CI-MPACB can take the place of multiport DCCB, which greatly reduce the construction cost of DCCB.

The proposed CI-MPACB topology is shown in Fig 2, which consists of a MBS and SBSs corresponding to the number of lines. The MBS consists of a current injection branch and a energy dissipation branch. The current injection branch is comprised of an inductor $L_H$, a high voltage capacitor $C_H$ and a square wave excitation source. The combination of precharged capacitor $C_P$ and IGCTs is designed as the wave excitation source. The SBS consists of a main breaker (MB) and an auxiliary breaker (AB).

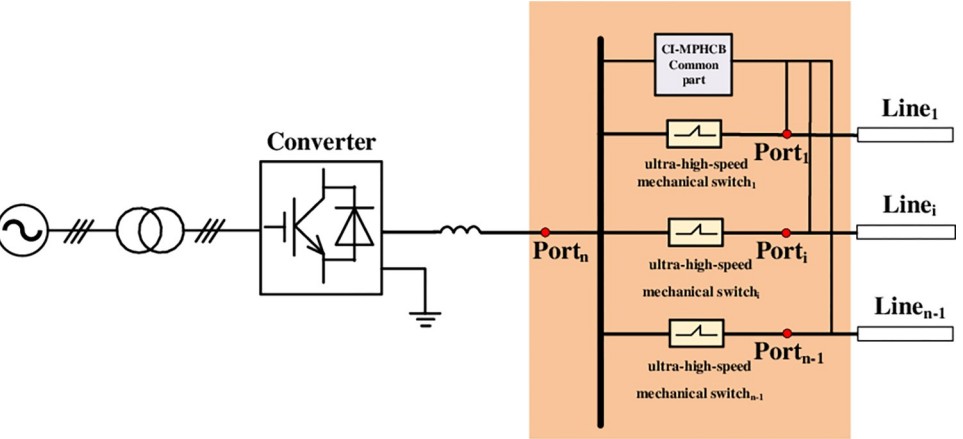

**Fig 1. The configuration of CI-MPACB.**

In Fig 2, $I_{dc}$ is the total DC current. $I_{MB}$ is the current of the MB. $I_C$ is the current of the current injection branch. $I_{MOV}$ is the current flowing through the energy dissipation branch. $U_E$ is the external voltage of the square wave excitation source, with a initial value equals to the precharge voltage $U_0$.

## 2.2 Principle of CI-MPACB fault-division control

As shown in Fig 3, it is the current path under normal operation of the line. When a fault occurs, the CI-MPACB will go through following stages during the breaking process. The opening sequence control logic in case of line$_i$ fault is given by:

Stage1: $t \leq t_{fault}$: Before the fault occurs, MBs of n lines are in the closing state, and ABs is in the opening state.

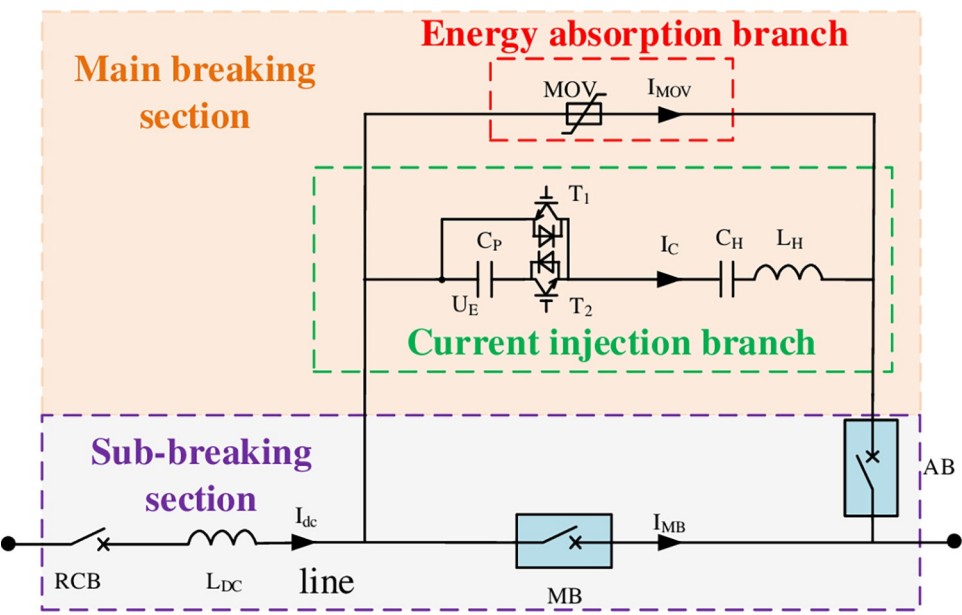

**Fig 2. Topology of the proposed CI-MPACB.**

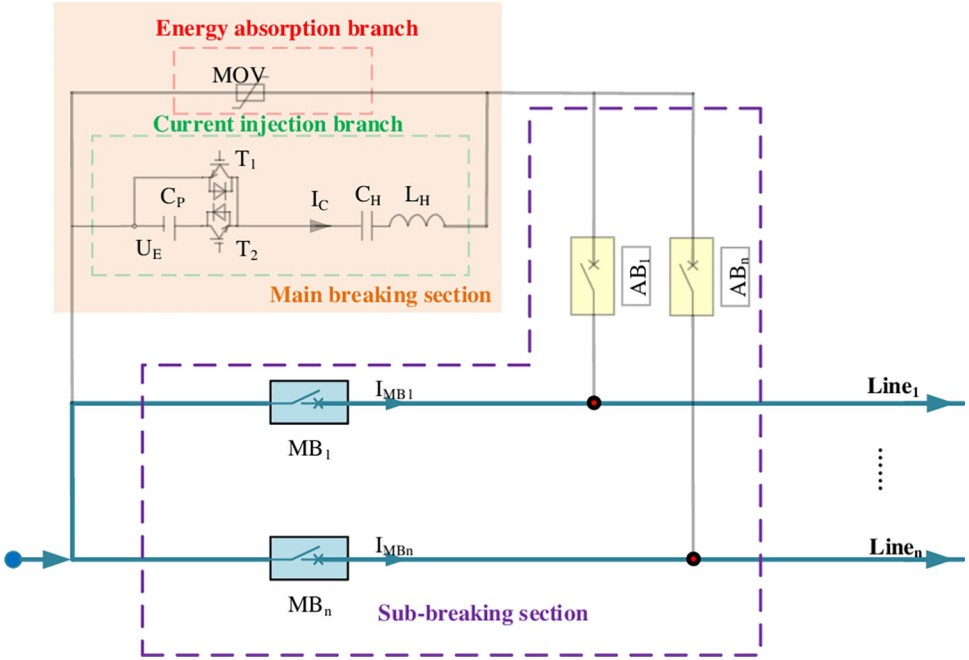

**Fig 3. Normal operation stage.**

Stage2: $t_{\text{fault}} \leq t \leq t_{\text{CI}}$: As shown in Fig 4(A), the fault occurs at the time instant $t_{\text{fault}}$. After receiving the fault removal command of line$_i$, the control device immediately give offs command to the AB$_i$ closing and MB$_i$ opening of line$_i$. The MB$_i$ should be opened and the AB$_i$ should be closed immediately. After detecting that the AB$_i$ is closed and MB$_i$ is opened in place within 2ms, start the electronic trigger switch of the MBS. The increasing current is generated in current injection branch. The process of Stage2 is shown in Fig 4(B).

Stage3: $t_{\text{CI}} \leq t \leq t_{\text{EA}}$: When the $I_{\text{MB}_i}$ crosses zero and the arc is extinguished, the $I_{\text{dc}}$ transfers to the MBS. When the voltage at the resonant capacitor end exceeds the action voltage of the MOV, the MOV consumes energy and absorbs the line breaking energy. Detect current of $I_{\text{dc}_i}$ to 0, lock the main electronic trigger switch. The process of Stage3 is shown in the Fig 4(C).

Stage4: $t_{\text{EA}} \leq t \leq t_{\text{off}}$: AB$_i$ should be broken. The fault line is disconnected at the time instant $t_{\text{off}}$.

The closing signal of the AB is an optical signal, and the communication protocol is consistent with the MB. Closing response time less than 50us

where $t_{\text{CI}}$ is the time when the current injection branch starts. $t_{\text{EA}}$ is the time when the energy absorbing branch starts.

Both MB and AB use high-performance epoxy resin to seal the pole, which has good vacuum sealing, insulation performance, resistance to high and low temperature impacts, and mechanical impact performance.

## 2.3 Operating principle of MBS

The waveforms related to the MBS breaking process are shown in Fig 5.

The following expound shows the full operation sequence:

1. $0$-$t_0$: In advance to the operation of the CB, the controllable electronics are blocked and the capacitor C$_P$ is pre-charged by an external DC power supply.

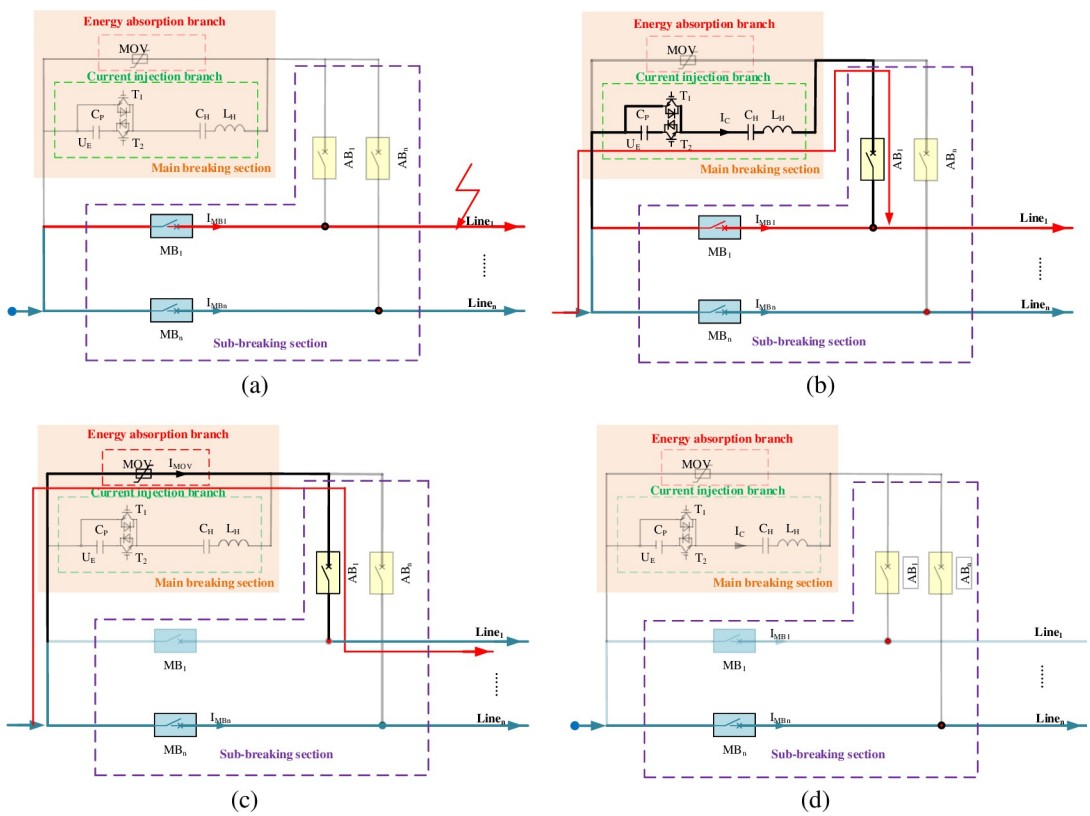

**Fig 4. Fault detection stage.** (a) Line fault occurs: t = tfault, (b) Current injection branch operates: tfault≤ t≤tCI, (c) Energy absorption branch operates: tCI≤ t≤ tEA, (d) Fault line is disconnected: t = toff.

2. $t_0$-$t_1$: The fault occurs at time $t_0$. Accordingly, the line current starts to increase, then the fault current limiting reactor (LDC) reduces the increasing rate of the line current. A trip signal is sent to the CI-MPACB at instant $t_1$.

3. $t_1$-$t_2$: As the CI-MPACB receives the trip signal at $t_1$, the MB starts to separate the contacts, which reach a sufficient gap distance at $t_2$.

4. $t_2$-$t_3$: At $t_2$, the current injection branch is activated. A zero-crossing of arc current is produced by the generation of the resonant current, which steadily grows in amplitude every half cycle.

5. $t_3$-$t_4$: $I_{MB}$ crosses zero under the superposition of $I_C$ and the arc is extinguished at $t_3$. $I_{dc}$ charges the resonant capacitor $C_H$ until MOV action voltage is reached.

6. $t_4$-$t_5$: The MOV consumes energy. Finally, complete the entire break process.

Fig 6 shows the topology model when $T_2$ and $T_1$ are switched. As shown in Fig 6(A), $T_2$ is triggered in the first cycle. Then the trigger signal is removed. $C_P$ continuously discharges externally and current injection branch generates $I_C$ flowing clockwise. And the current gradually increase while charge $C_H$. $U_{C_H}$ also increases. When the $I_C$ reaches the clockwise peak, the change rate of the $I_C$ is zero, which means that the $U_{L_H}$ at this time drop across zero. According to Kirchhoff's voltage law, $U_{C_H} = U_0$.

In Fig 6(B), the amplitude of the $I_C$ decreases in the clockwise direction due to the continuous current action of inductor $L_H$. It still charges $C_H$. When $I_C$ drops to zero, the change rate

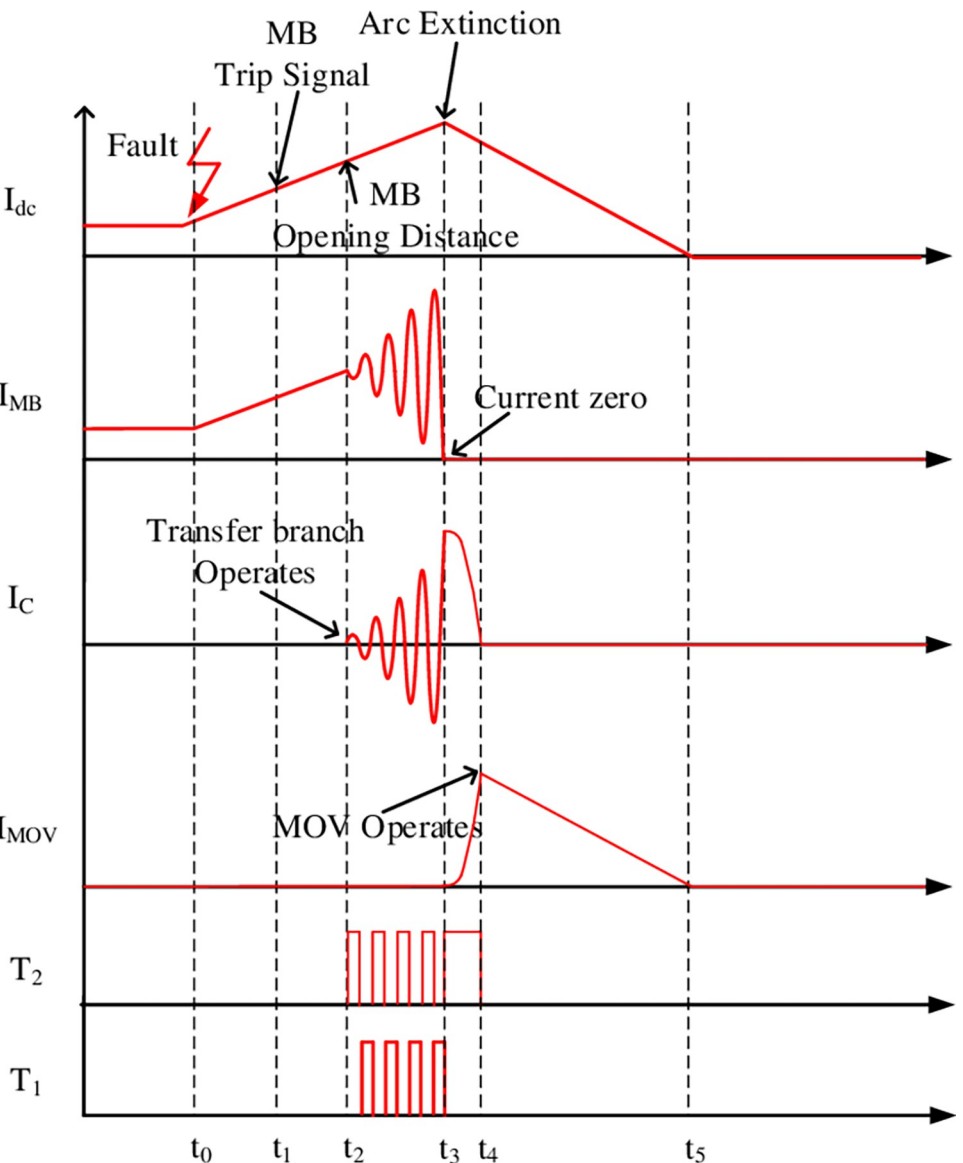

**Fig 5. Waveforms of MBS.**

of $I_C$ is the largest, and the $U_{L_H}$ gets its maximum. At this time, $U_{C_H}$ reaches the first peak value. And the $I_C$ changes direction across zero.

In Fig 6(C), when $T_1$ is triggered, the $I_C$ will increase in the counterclockwise direction after crossing zero. When the $I_C$ reaches the counterclockwise peak, the polarity of the $U_{C_H}$ will change.

In Fig 6(D), the $I_C$ drops from the counterclockwise peak until the $U_{C_H}$ reaches its peak.

## 3 Electrical parameter design

### 3.1 Circuit analysis of the MBS

When the MBS of CI-MPACB starts to operate at t2, the initial voltage of pre-charged capacitor Cp is U0 and UCH is zero when T2 is firstly opened in the current injection branch. In the

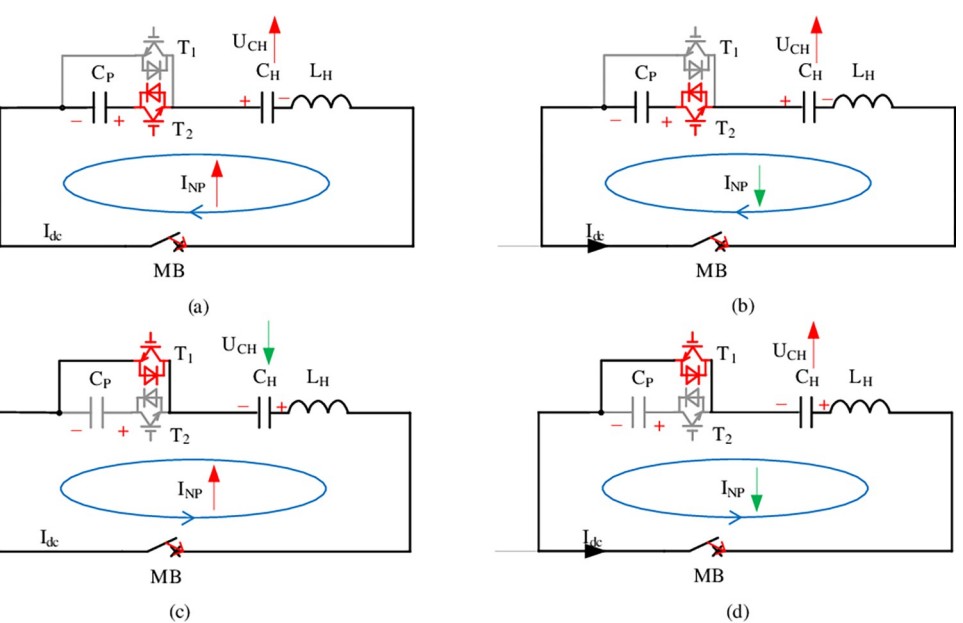

**Fig 6. The generation of IC and the conduction path of IGCTs.**

equivalent circuit, CH is serial connected with Cp, accordingly the equivalent capacitance of the resonant circuit can be calculated by (1). The total equivalent capacitance is approximately equal to CH because of Cp≫CH. The resonance angular frequency (ω) is obtained from (2), and the natural frequency of the commutation branch is fLC (3).

The following is a detailed description and formula expression of the t2-t3 process mentioned above. In the first stage of the first resonant period (T2 turn on, T1 turn off), the resonance of the first half-cycle current ($I_{1_{T2}}$) is a second-order oscillatory current discharge process from the excitation source U0 of the Cp to CH and LH. In the second stage (T2 turn off, T1 turn on), there is no pre-charge capacitor Cp in the equivalent circuit, so the resonant current $I_{1_{T1}}$ is generated under the action of CH and LH. In the first stage of the m-th resonant period (T2 turn on, T1 turn off), In the equivalent circuit, CH and Cp are linked in series. At this point when the T2 conduction, the $I_{m_{T2}}$ can be regarded as the second-order resonant current of the excitation source UE charging and discharging LH formed by CH and Cp in series.

In the second stage (T2 turn off, T1 turn on), the $I_{m_{T1}}$ can be regarded as the two-order resonant current generated by CH and LH. When the fault current (If) is interrupted at tI, then zero-crossing is generated at tI(4).

$$C_{OSC} = \frac{C_p C_H}{C_p + C_H} \approx C_H \tag{1}$$

$$\omega = \frac{1}{\sqrt{C_H L_H}} \tag{2}$$

$$f_{LC} = \frac{1}{2\pi\sqrt{C_{HL}}} \tag{3}$$

$$I_f - I_{P_{Ti}}(t_I) = 0, \ (I_{1_{T2}}, \ I_{1_{T1}}, \ I_{m_{T2}}, \ I_{m_{T2}} \in I_{P_{Ti}}) \tag{4}$$

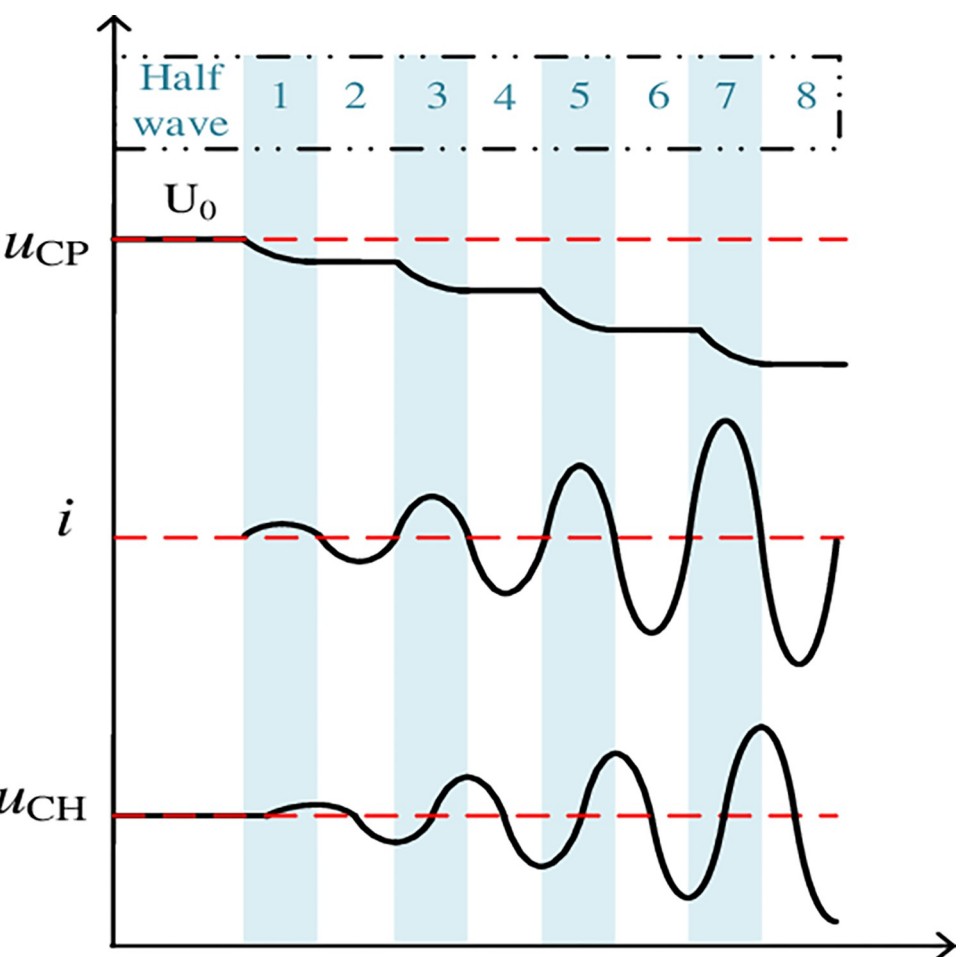

**Fig 7. Schematic diagram of MBS electrical quantity.**

The resonant current $I_C$ determines the commutation capacity of CI-MPACB. The $I_C$ of CI-MPACB in this design is different from that of traditional mechanical DCCB. Its essence is a resonant current with increasing amplitude. The typical waveform is shown in Fig 7. To describe its commutation characteristics accurately, the process of resonant current $I_C$ generation will be used to clarify the mathematical model of MBS commutation and serve as the basis for parameter design.

Fig 8(A) shows the topology of the complex frequency domain equivalent circuit when $T_2$ is turned on for the first time Fig 8(A). The formula of this process in the complex frequency domain is listed by using the loop current method:

$$I(s)\left(\frac{1}{sC_P} + \frac{1}{sC_H} + sL_H\right) = -\frac{U_{C_P}(t_{0-})}{s} \tag{5}$$

After $I(s)$ is resolved, the expression of the release current $i_{1_{T2}}(t)$ of the resonant circuit is obtained after inverse Laplace transform:

$$i_{1_{T2}}(t) = \frac{\sqrt{C_H}\sqrt{C_P}U_0\sin(\sigma_1)}{\sqrt{L_H}\sqrt{C_H + C_P}} \tag{6}$$

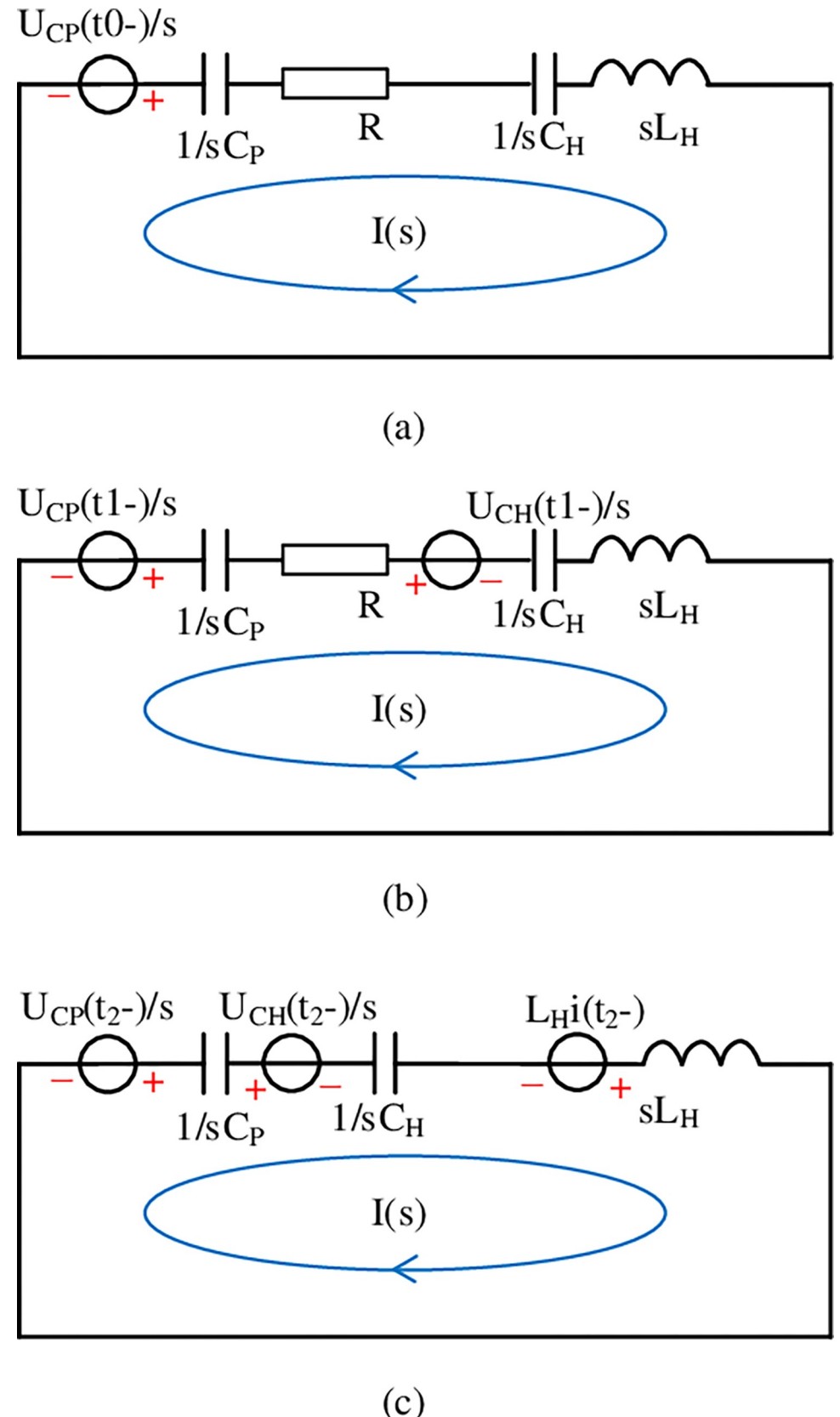

**Fig 8. Complex frequency domain equivalent circuit.**

$$U_{C_{H-1}T_2}(t) = U_{0^-} - \frac{C_P U_0 \cos(\sigma_1)}{C_H + C_P} \tag{7}$$

$$U_{C_{P\_1}T_2}(t) = U_0 + U_0 \cos(\sigma_1) - \frac{C_P U_0 \cos(\sigma_1)}{C_H + C_P} \tag{8}$$

where $\sigma_1 = \frac{t\sqrt{C_H + C_P}}{\sqrt{C_H}\sqrt{C_P}\sqrt{L_H}}$

Fig 8(B) is the complex frequency domain equivalent circuit topology diagram when $T_1$ is turned on for the m-th (m≥1) time. The formula of this process in the complex frequency domain is listed by using the loop current method:

$$I(s)\left(\frac{1}{sC_P} + sL_H\right) = \frac{U_{C_H}(t_{1_{T1-}})}{s} - L_H i\left(t_{T_{T-}}\right) \tag{9}$$

After $I(s)$ is found, the expression of the release current $i_{m_{T1}}(t)$ of the resonant circuit is obtained after the pull type inverse transformation:

$$i_{m_{T1}}(t) = C_H \left( \frac{\frac{U_{C_H}(t_{1_{T1-}})e^{\sigma_2}\sqrt{-C_H L_H}-}{C_H L_H}}{\frac{U_{C_H}(t_{1_{T1-}})e^{-\sigma_2}(e^{2\sigma_2+1})\sqrt{-C_H L_H}}{2C_H L_H}} \right) \tag{10}$$

$$U_{C_{H\_m_{T1}}}(t) = \frac{U_{C_H}(t_{1_{T1-}})e^{-\sigma_2}(e^{2\sigma_2+1})}{2} \tag{11}$$

where $\sigma_2 = \frac{t\sqrt{-C_H L_H}}{C_H L_H}$

Fig 8(C) shows the topology of the complex frequency domain equivalent circuit when $T_2$ is turned on for the n-th (n≥2) time. The formula of this process in the complex frequency domain is listed by using the loop current method:

$$I(s)\left(\frac{1}{sC_P} + \frac{1}{sC_H} + sL_H+\right) + \frac{U_{C_P}(t_{n_{T2-}})}{s} - \frac{U_{C_H}(t_{n_{T2-}})}{s} = 0 \tag{12}$$

After $I(s)$ is sorted out, the expression of the release current $i_{n_{T2}}(t)$ of the resonant circuit is obtained after inverse Laplace transform:

$$i_{n_{T2}}(t) = -\frac{\sqrt{C_H}\sqrt{C_P}\sin(\sigma_1)(U_{C_H}(t_{n_{T2-}}) - U_{C_P}(t_{n_{T2-}}))}{\sqrt{L_H}\sqrt{C_H + CP}} \tag{13}$$

$$U_{C_{H\_n_{T2}}}(t) = \frac{C_P \cos(\sigma_1)(U_{C_H}(t_{n_{T2-}}) - U_{C_P}(t_{n_{T2}}))}{C_H + C_P} + U_{C_P}(t_{n_{T2-}}) \tag{14}$$

$$U_{C_{P_n T2}}(t) = U_{C_P}(t_{n_{T2.}}) - \frac{C_H \cos(\sigma_1)(U_{C_H}(t_{n_{T2}}) - U_{C_P}(t_{n_{T2-}}))}{C_H + C_P} \tag{15}$$

## 3.2 Method of parameter design based on multiple constraints and cost optimization

In order to find the best combination of $C_H$, $C_p$, $U_0$, and $L_H$ for the current injection branch, and meet the requirements for fast and reliable breaking of the MBS, it is necessary to optimize the parameters of the device. So as to realize the compactness and low cost of MBS. Therefore, the following five constraints are given in combination with the commutation characteristics and interrupting process of MBS.

Constraint 1: To ensure reliable commutation of DC current, the peak value of resonant current shall be greater than the maximum value of DC current to be disconnected.

$$I_{C\_max} > I_{dc\_max} \tag{16}$$

Conestraint 2: In order to realize fast disconnection, the corresponding cut-off time of $t_1 \sim t_3$ should be less than 3ms.

Conestraint 3: In order to prevent MB from re breakdown, the growth rate of $U_{CH}$ should be less than the MB medium strength $U_{MB\_d}$ recovery speed in $t_3 \sim t_4$ stages.

$$d(U_{CH})/dt = I_{dc}/C_H < d(U_{MB\ d})/dt \tag{17}$$

Conestraint 4: The upper and lower limits of resonance frequency are generally 3 kHz to 10 kHz according to [31]. The current slope at current zero and the selection of $C_H$ and $L_H$ parameters are determined by the resonance frequency.

$$3 \times 10^3 \le f_{LC} \le 10^4 \tag{18}$$

Conestraint 5: In order to reduce the voltage withstand requirement of components, $C_p$ shall be ensured have no obvious overvoltage. It be obtained according to the voltage division principle of series capacitor.

$$U_{C_p} = U_{MOV}C_H/(C_H + C_p) < kU_0 \tag{19}$$

where, $U_{C_p}$ is the voltage $C_p$ bears during transient breaking. $U_{MOV}$ is the highest voltage of MOV. $k$ is the reliability coefficient.

For the determined and $C_H$, $C_p$ and other parameters have array values, corresponding to different costs, the minimum total cost is taken as the optimization goal.

To reduce the volume and cost of DCCB, its total capacitance $W_s$ should be as small as possible:

$$W_S = (1/2)C_H U_{MOV}^2 + (1/2)C_p(kU_0)^2 \tag{20}$$

## 3.3 Case study of parameter design

Taking DCCB with rated voltage $U_{rate}$ of 50 kV, rated current $I_{rate}$ of 2 kA and maximum breaking current $I_{dc\_max}$ of 25 kA as an example. The action time of MB is 3 ms. According to Eq (16), $I_C$ can reach 25 kA within 1ms at most.

$I_{rate}$ corresponds to the slowest condition of $C_H$ charging. And the maximum value of $C_H$ is 40 uF (corresponding to Constraint 2).

Since the dielectric strength recovery rate of MB is about 1 kV/μs [34]. Considering $I_{dc\_max}$ is the fastest charging condition of CH, the minimum value of $C_H$ is 25 μF according to (17) (corresponding to Constraint 3).

Since it is known that CH is not less than 25 μF. According to Eq (3) ($f_{LC}$) and the Constraint 4, $L_H$ should be greater than 7.2 μH and less than 80.4 μH. The stray inductance of the

existing 50 kV voltage level DCCB is about 46 µH. In order to reduce the difficulty of commutation, $L_H$ is taken as stray inductance.

The voltage ratio of MOV is about 1.6 times [32]. According to Eq (19), when $C_H$, $U_0$ and k are determined,$C_p$ limit value can be determined.

Starting from the minimum value of $C_H$ as 25 µF, set k as 1.2 to obtain the optimal values of $C_p$ and $U_0$ when $C_H$ changes. According to the research above, a commutation capability of at least 25 kA is needed to guarantee a dependable interruption fault current. According to (20), higher capacitor pre-charged voltage will result in larger bulk and higher cost. Therefore, the parameters of the current injection branch are recommended as follow, $L_H$ = 46 µH, $C_H$ = 35 µF, $C_p$ = 4000 µF, and $U_0$ = 4 kV.

## 4 Scheme comparison and cost analysis

### 4.1 Technical comparison of circuit breakers

To implement the fault current interruption, the proposed topology makes use of a commutation branch made up of LC and IGCT components. Table 1 compares the performance of the DCCB with traditional interruption technologies.

### 4.2 Comparison of costs

Additionally, the cost comparisons between the planned and current 50 kV ratings results are provided in Table 2. The design and selection of MOV must take a lot of redundancy into account in order to ensure the availability of the fault energy dissipation. Because this decision is largely influenced by the system parameters, further optimization is challenging. Therefore, the primary factor is the price of PE devices.

1) Topology of Scheme 1

As for scheme 1, there are m load commutation switches (LCSs), m MBS in scheme 1. The IGBT (3000A4500V) module is used for the LCS and MB.

2) Topology of Scheme 2

Only one MBS is required in Scheme 2 because to the H-bridge structure. However, one DC line now has two LCSs linked to it instead of just one. In total, there are 2×m LCSs, one MBS in scheme 2.

3) Proposed topology

A total of 1 MBS and m SBS need to be configured. The total number of IGCTs required for MBS is 12. And in the SBS, m MBs and ABs are required.

Compared with Scheme 1, the proposed CI-MPACB saves ¥(375×m-480) thousand on the cost of PE. When m = 2, the cost saving is ¥2.7million. When m = 3, the cost saving is ¥6.45 million. The proposed CI-MPACB saves ¥(150×m-180) thousand on the cost of PE compared to Scheme 2. When m = 2, the cost saving is ¥1.2million. When m = 3, the cost saving is ¥2.7

**Table 1. Technical comparison of CBs.**

| Parameters | Mechanical circuit breaker | Hybrid circuit breaker | CI-MPACB |
|---|---|---|---|
| On state power loss ($P_{T(AV)}$) | micro ohm, low | milliohm, high | micro ohm, low |
| Breaking capacity | rest with the arc extinguishing capacity of CBs | limited by PE turning off capacity | depends on the arcing out of the CBs and the current resistance of the PE |
| Commutation reliability | single zero crossing point of current, medium | high impedance forced power transfer, high | multiple zero crossing point of current, high |
| Energy supply system | high voltage and large capacitance capacitor bank | high pressure valve group composed of a large number of full control devices in series | few low-voltage submodule DC capacitors |

**Table 2. Price comparison of PE in CBs.**

| Item | Scheme 1 [33] | Scheme 2 [34] | | Proposed topology (IGCTs) |
|---|---|---|---|---|
| Breaking Capability | 50 kV/25 kA | 50 kV/25 kA | | 50 kV/25 kA |
| MBS/ SBS | MBS | MBS | SBS | MBS SBS |
| Required number | m | 1 m | | 1 m |
| PE | IGBT×15 | IGBT ×12 | IGBT ×6 | IGCT×12 / |
| Total cost | $C_{IGBT}×15×m$ | $C_{IGBT}×12+C_{IGBT}×6×m$ | | $C_{IGCT}×12$ |

$C_{IGBT}$: ¥25000, the price of IGBT [35].

$C_{IGCT}$: ¥40000, the price of IGCT.

million. According to the above analysis, it is obvious that under the current situation of DC power grid with multi outlet network structure, the proposed CI-MPACB can greatly reduce costs on the premise of ensuring reliability.

## 5 Case study of CI-MPACB in the VSC-HVDC

### 5.1 Operating waveform of CI-MPACB

As shown in Fig 10, the mathematical model is basically consistent with the PSCAD/EMTDC simulation results, which can verify the current transfer mathematical model and the best parameters in Fig 9. And according to Fig 10 (B) $I_C$ can reach 25 kA in 1ms.

In Fig 11(A), fault occurs at t0 = 0.5 s; When $t_1$ = 0.5120 s, the dynamic and static contacts of MB in the fault line are separated and arcing occurs and AB in the fault line starts to close;

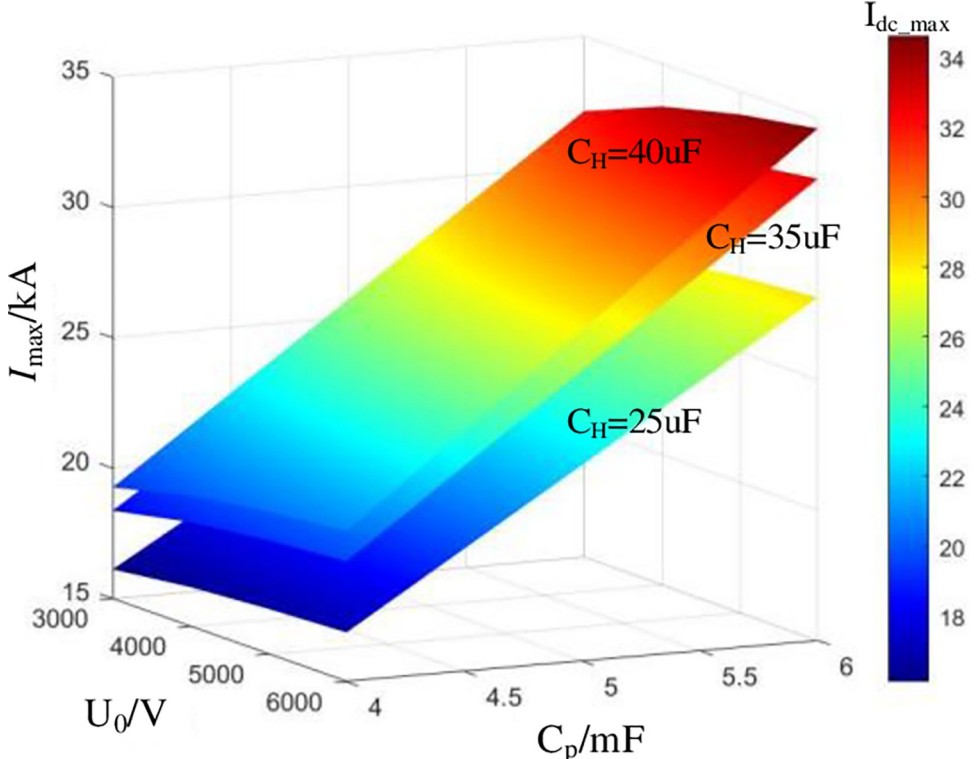

**Fig 9. Optimization results of current injection branch parameters.**

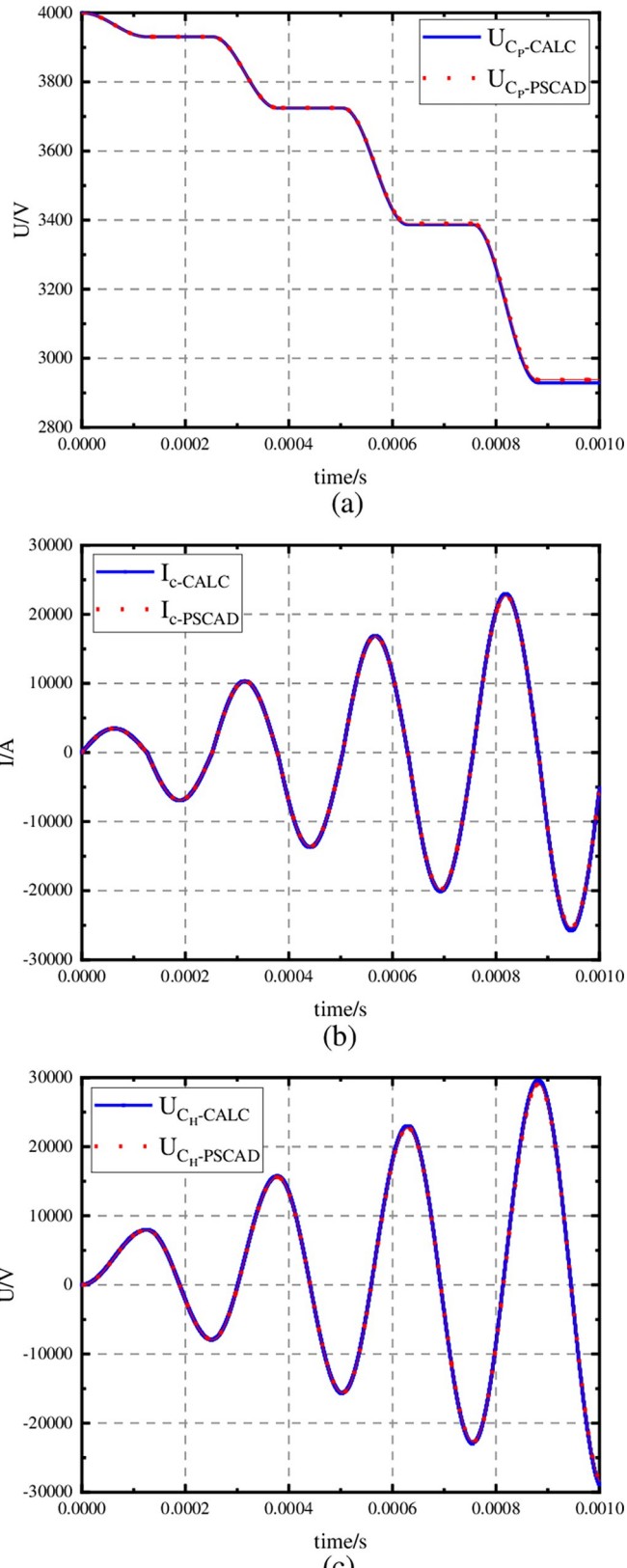

**Fig 10. Verification of the correctness of the design of the parameters of the current transfer mathematical model.**
(a) Comparison results of $U_{C_P}$, (b) Comparison results of $I_C$. (c) Comparison results of $U_{C_H}$.

When $t_2$ = 0.5140 s, AB is closed. Then the MBS of the CI-MPACB takes turns IGCT according to the inherent frequency of the commutating branch. When $t_3$ = 0.5147 s, $I_{dc}$ cross zero and transfer to the commutation branch. When $t_4$ = 0.5149 s, the voltage of the converter branch reaches the MOV reference voltage. Then the MOV is connected and forced to $I_{dc}$ drops. When $t_5$ = 0.5265 s, $I_{dc}$ is disconnected by residual current switch when it is close to zero.

As shown in Fig 11(B), residual voltage $U_{MOV}$ of 50 kV DCCB is 80 kV, which is sufficient to attenuate $I_{dc}$. From the $U_{CH}$ waveform, it oscillates and rises under the charge of the square wave excitation source. This is because that $I_C$ can increase the cause of resonance.

As shown in Fig 12, the current superposition of T1 and T2 tubes is the current of the current injection branch. And the voltage amplitude of T1 and T2 tubes decreases with the increase of turn-on times.

## 5.2 Simulation of CI-MPACB

In order to prove the feasibility of the CI-MPACB, a three terminal DC system is used (Fig 13).

As shown in Fig 13, a simulation model with three terminal VSC-HVDC system is built in PSCAD/EMTDC. The voltage source converter is mainly consisted of fully controlled circulating current bridge, DC side capacitor, AC measurement converter transformer or converter reactor, and AC filter. Three-phase and two-level architecture is adopted in the fully controlled converter bridge. And each bridge arm is composed of multiple IGBT or GTO and other turnoff devices. The capacitor in DC side supports the converter's voltage and buffers the inrush current when the bridge arm is turned off and reduce harmonics in DC side. The function of the AC measurement converter transformer or converter reactor is to filtrate harmonics on AC side. Transmission lines are transmitted by overhead lines, and Bergeron model is used to describe transmission lines. The parameters of the system are shown in Table 3.

Assume t = 0.5s on the transmission line12, the fault occurs at a distance of x(x$\leq$200) km from S1. As shown in Fig 14(A), the system remains steady state, and load current IMBline12 of SBS in CI-MPACB1flow is unchanged from 0s to 0.5s. After the fault occurs, IMBline12 starts to rise rapidly. When t = 0.501 s, the dynamic and static contacts of MBline12 are separated and arcing occurs. Then the ABline12 starts to close for 2ms. When the ABline12 closure is completed, the MBS of the CI-MPACB rotates IGCT according to the inherent frequency of the current injection branch. As the resonant current increases, the fault current is rapidly transferred to the resonant branch. When the Idcline12 charges capacitor CH to the triggering conduction voltage of the MOV, Idcline12 then transfer to the energy consuming branch immediately. Idcline12 gradually decreases to 0. Under three fault conditions, the rising rate and amplitude of Idcline12 is decreased with the increase of distance. When x = 25, the peak value of Idcline12 is 23.38 kA. When x = 50, the peak value of Idcline12 is 17.96 kA. When x = 100, the peak value of Idcline12 is 12.14 kA.

As shown in Fig 14(B), the time of current injection into SBS decreases as the fault distance reduces (i.e., the number of resonant cycles increases). When $x$ = 25, the time when the value of $I_C$ reaches the maximum value is 0.51506 s. When $x$ = 50, the time when the value of $I_C$ reaches the maximum value is 0.51480 s. When $x$ = 100, the value of reaches $I_C$ the maximum value is 0.51453 s.

As shown in Fig 14(A) and 14(C), the time at which MOV is turned on and forces $I_{dcline12}$ down to zero increases with the distance to the fault. The decreasing rate of $I_{MOV}$ decreases with the fault distance increasing. At $x$ = 25, the value of $I_{MOV}$ reaches zero in 0.5228 s, at $x$ = 50, the value of $I_{MOV}$ reaches zero in 0.5268 s, and at $x$ = 100, the value of $I_{MOV}$ reaches zero in 0.5315 s.

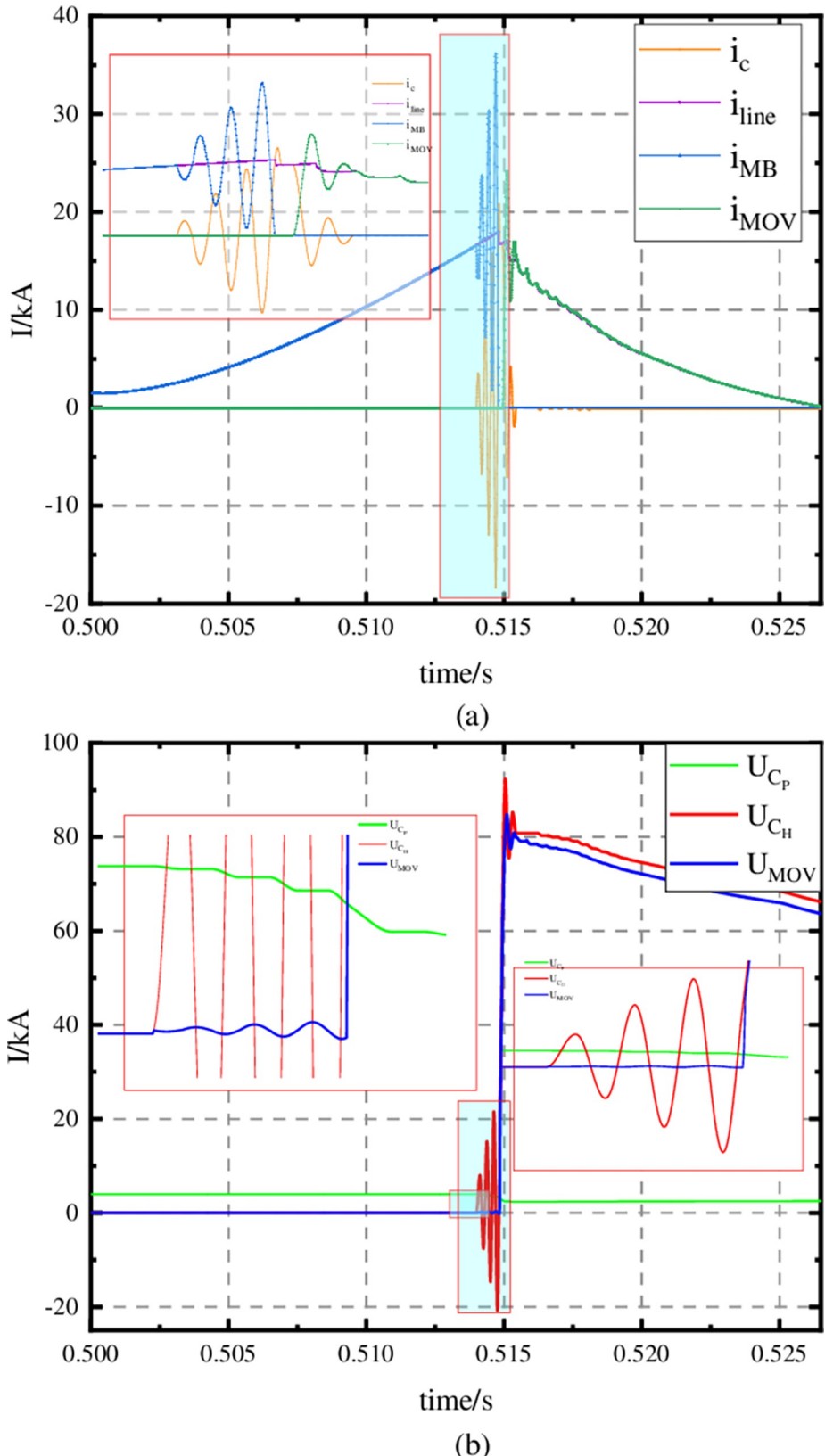

**Fig 11. Fault break waveforms.** (a) waveforms of current, (b) waveforms of voltage.

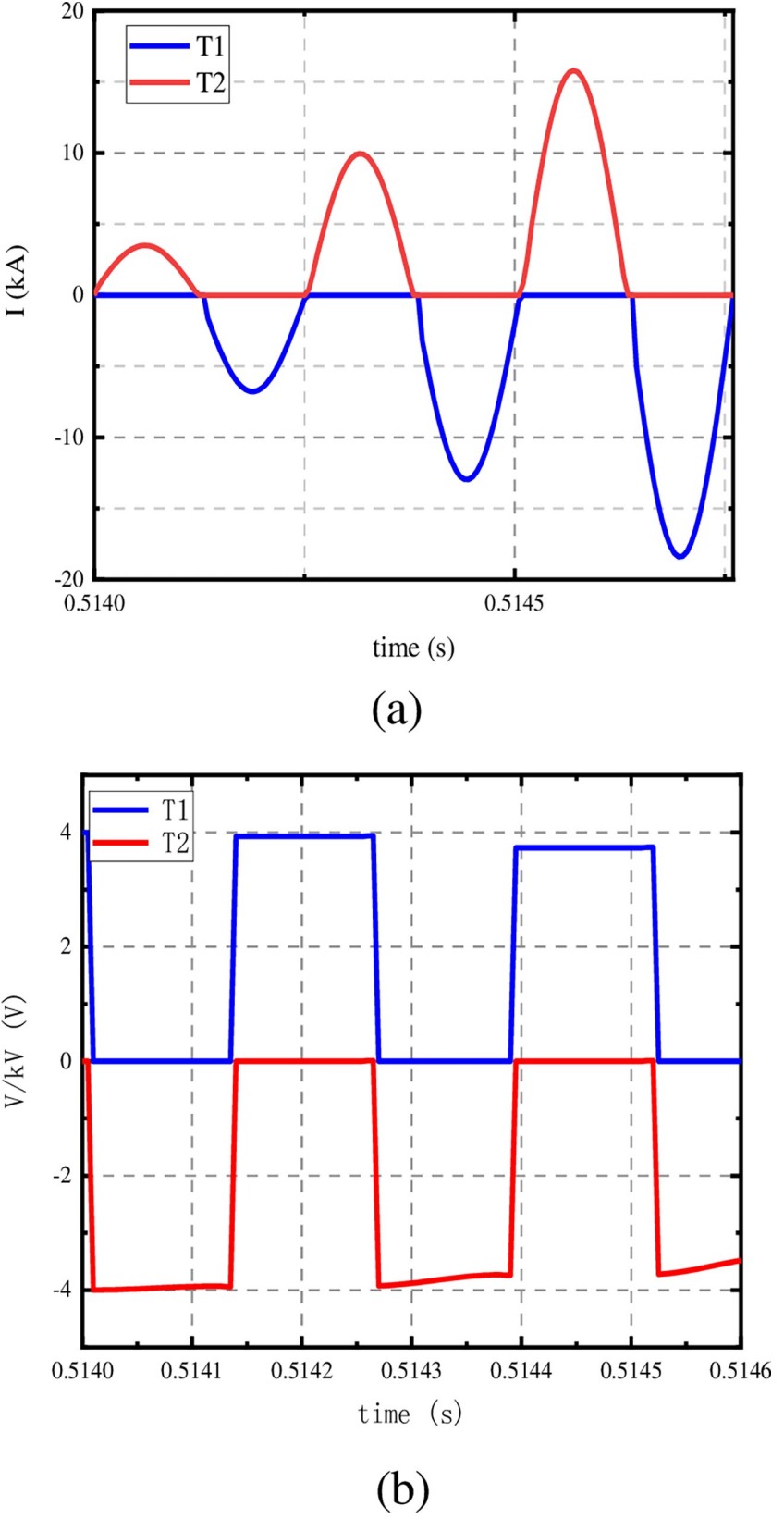

**Fig 12. T1 and T2 electrical quantity waveforms.** (a) waveforms of current, (b) waveforms of voltage.

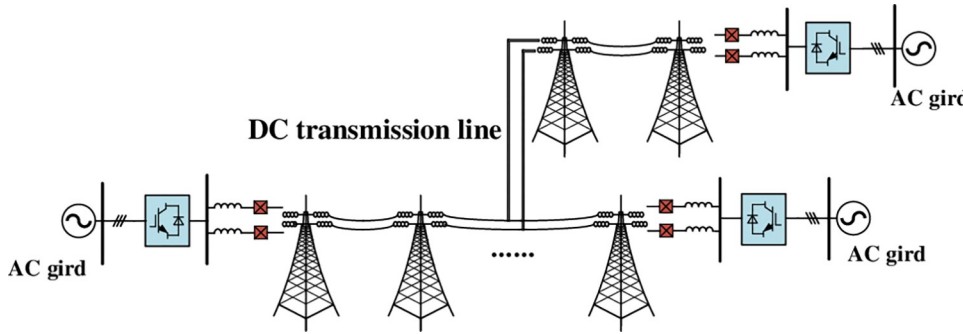

**Fig 13. The three-terminal DC system.**

As shown in Fig 14(D), the amplitude of IMB decreases with the increasing distance to the fault. At x = 25, the amplitude of IMB is 45.48 kA. At x = 50, the amplitude of IMB is 36.16 kA. And at x = 100, the amplitude of IMB is 24.84 kA.

As shown in Fig 15(A), the decreasing rate and value of the pre-charged capacitor voltage $U_{CP}$ increases with the fault distance increasing. At x = 25, the value of $U_{CP}$ decreases to 2.858 kV, at x = 50, the value of $U_{CP}$ decreases to 2.433 kV, and at x = 100, the value of $U_{CP}$ decreases to 2.029 kV.

As shown in Fig 15(B), the time to reach the peak of the resonant capacitor voltage $U_{CH}$ decreases with the fault distance increasing, and the peak value also decreases with the fault distance increasing. At x = 25, the peak value of $U_{CH}$ is 92.286 kV, at x = 50, the peak value of $U_{CH}$ is 90.189 kV, and at x = 100, the peak value of $U_{CH}$ is 85.032 kV.

As shown in Fig 15(C), the time to reach the peak of MOV voltage $U_{MOV}$ and the peak value decreases with the fault distance increasing. At x = 25, the peak value of $U_{MOV}$ is 84.239 kV, at x = 50, the peak value of $U_{MOV}$ is 83.762 kV, and at x = 100, the peak value of $U_{MOV}$ is 79.247 kV.

A comparison of the line currents $I_{dcline12}$ and $I_{dcline13}$ in the three cases is shown in Fig 16. When the line between S1 and S2 is faulty, the CI-MPACB can immediately open the faulty line, and the line between S1 and S3 operates normally.

The CI-MPACB was applied to the ± 160 kV Nan' ao multi terminal flexible DC transmission system. And compared with the MCB designed in paper [36]. Adopting a circuit breaker scheme with three 50kV sub-modules connected in series. The parameters of each sub-modules are selected as the optimization value mentioned above. The fault breaking waveforms are shown in the Fig 17.

As shown in Fig 17, the MB of the MCB proposed in the paper [36] starts to open and arc at the moment of 15 ms, and the current flowing through the MB continues to rise. The breaking current of the mechanical switch rises to 10.3kA at 18.5 ms. The commutator branch of the

**Table 3. Configuration parameter of a three-terminal DC grid.**

|  | **Item** | **Value** |
|---|---|---|
| DC side | DC rated voltage | ±50 kV |
|  | Length of dc line12 | 200 km |
|  | Length of dc line13 | 200 km |
|  | Length of dc line23 | 150 km |
| AC side | AC system voltage(L-L, RMS) | 23 kV |
|  | AC system inductance | 6 mH |
|  | AC system resistance | 0.1 Ω |

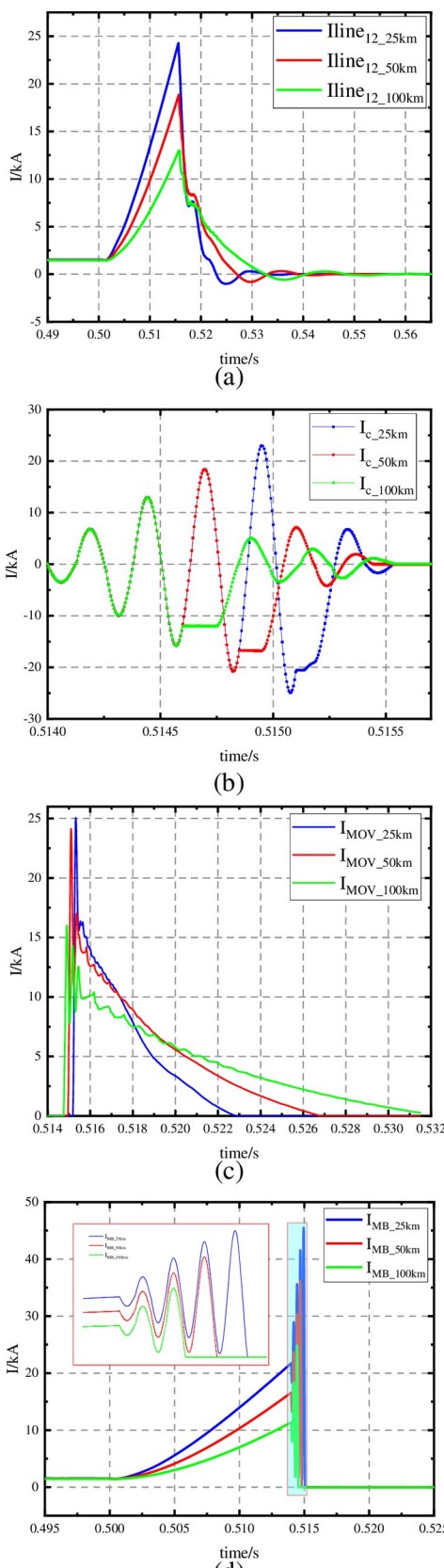

**Fig 14. Dynamic performance of the HVDC breaker (1).** (a) Currents of line $I_{cline12}$. (b) Currents of current injection branch $I_C$. (c)Currents of energy absorption branch $I_{MOCV}$. (d) Currents of SBS section $I_{MBline12}$.

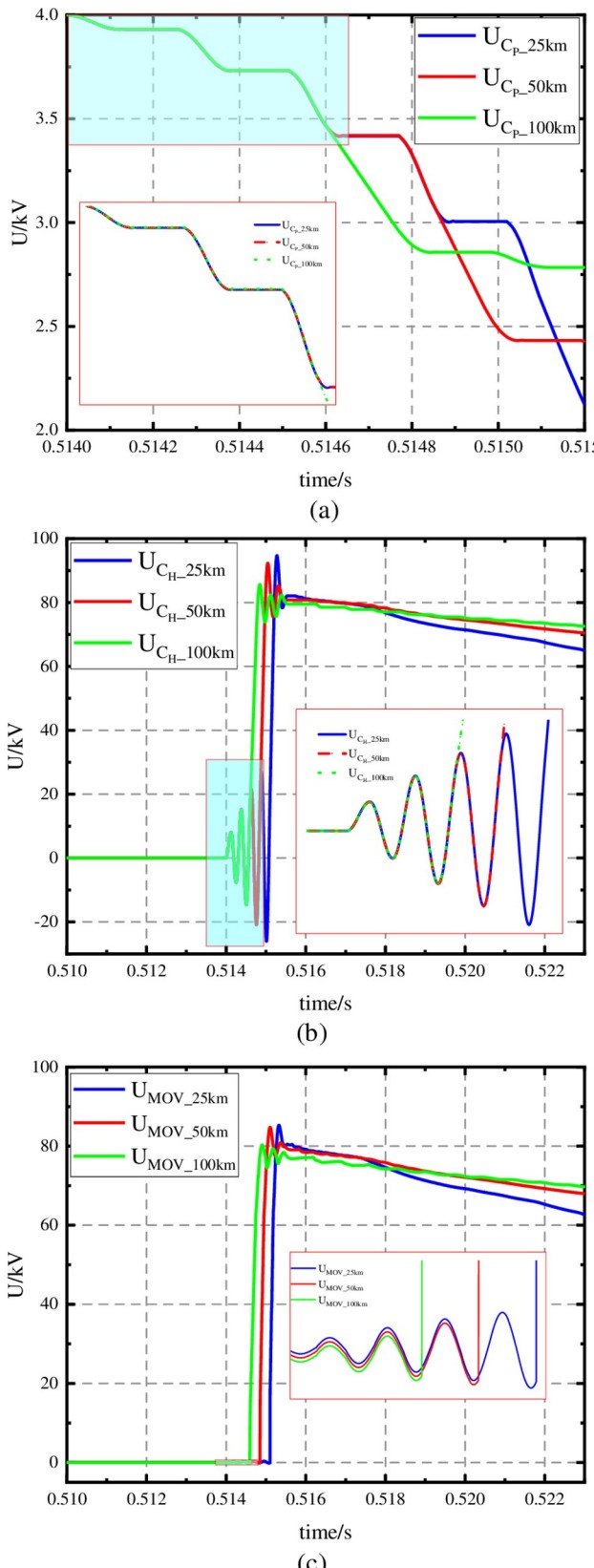

**Fig 15. Dynamic performance of the HVDC breaker (2).** (a) Voltage of $C_P$: $U_{CP}$. (b) Voltage of $C_H$: $U_{CH}$. (c) Voltage of $C_H$: $U_{MOV}$.

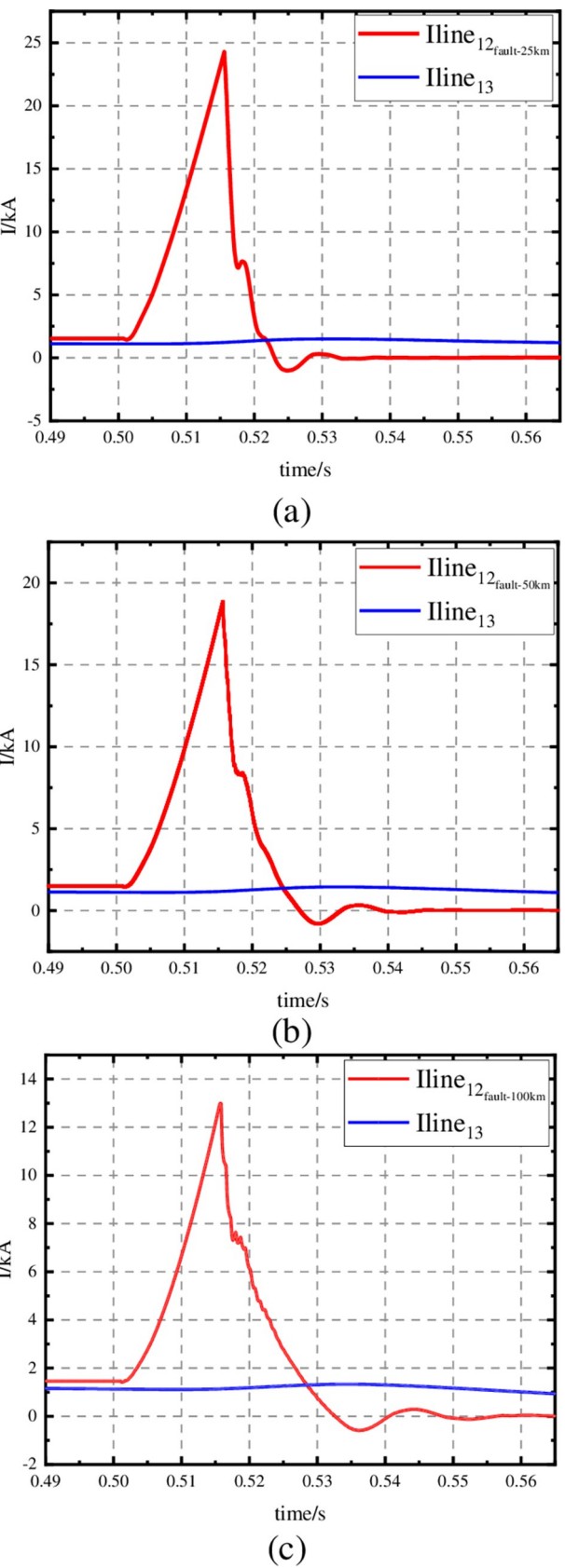

(a)

(b)

(c)

**Fig 16. Current waveforms of relevant line.** (a) Current waveforms: Iline12-25km and Iline13, (b) Current waveforms: Iline12-50km and Iline13, (c) Current waveforms: Iline12-100km and Iline13.

MCB is switched on. The precharge capacitor discharges and injects oscillating current into the MB. At the time of 18.89 ms, the current of MB oscillates to zero. The breaking time is 3.89ms. The peak value of superimposed current is 22.7kA. Under the same short-circuit condition, the MB of CI-MPACB starts to open at 16.5 ms. The current iniection branch is switched on at 18.5ms. The precharge capacitor discharges and T1 and T2 turn on alternately. At 18.8ms, the current of MB oscillates to zero and completes the breaking. The breaking time is 2.8ms. The peak superposition current is 20.07kA. The breaking time of CI-ACB is smaller than that of MCB. And the current stress is smaller.

Therefore, the CI-MPACB has a fast breaking speed and the injection current with an increasing periodic amplitude can form multiple zero crossing points, which can adapt to various short-circuit currents. This circuit breaker has high reliability in practical system applications.

## 6 Conclusion

A novel CI-MPACB for VSC based DC system is proposed in this paper. Its operating principle is outlined in detailed. A three-terminal HVDC grid with CI-MPACB is simulated in PSCAD/EMTDC, the validity and the feasibility are proved according to simulation results, and following conclusions could be drawn.

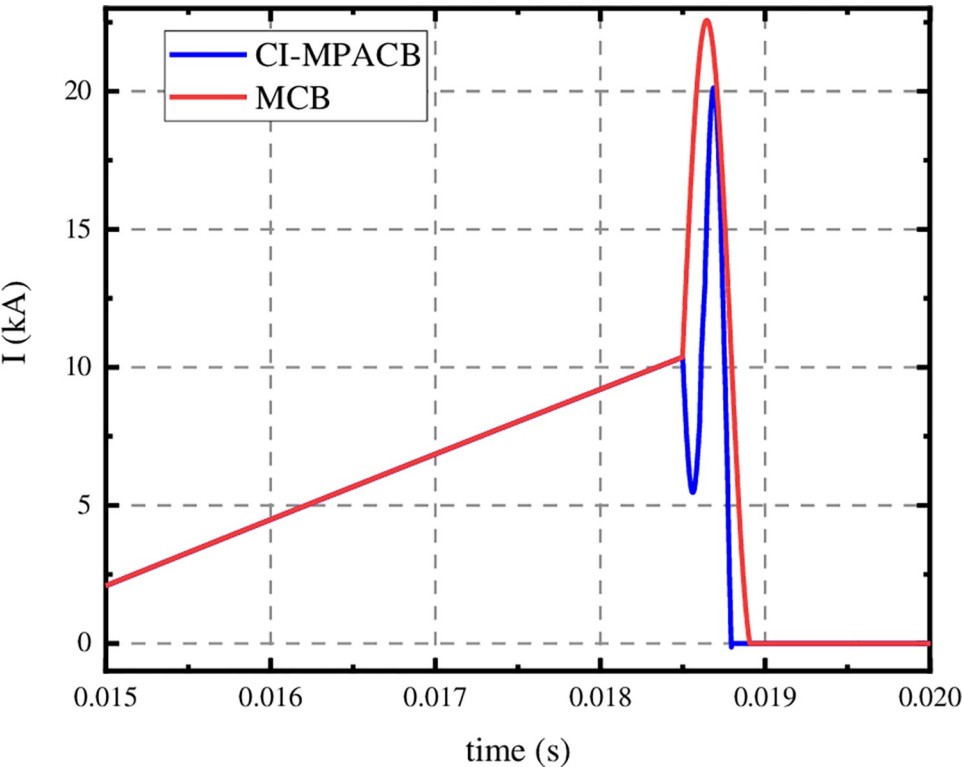

**Fig 17. Current waveforms of MB.**

1. The controlled resonant commutation principle of the MBS of the CI-MPACB creates multiple arc extinction zeros for the rapid mechanical switch. At the same time, the voltage level of power electronic devices can reduce to 10% of the maximum transient breaking voltage, and the breaking current is close to zero.

2. The CI-MPACB adopts the design idea of ADCCBs. One current injection branch and energy dissipation branch are shared by every lines connected to the same DC bus. This feature fits well for HVDC applications.

3. The CI-MPACB not only reduces the number and cost of power electronics devices by designing the resonant circuit in the transfer branch, uses IGCT to replace IGBT as the conduction device of the resonant circuit. It eliminates the voltage instability caused by a large number of IGBTs in series and further reduces the cost.

There will be an increasing number of multiport HVDC networks in the future. These applications stand to gain greatly from the proposed CI-MPACB.

## Author Contributions

**Conceptualization:** Puyi Cui.

**Data curation:** Guoli Li.

**Formal analysis:** Qian Zhang.

**Methodology:** Qinglian He.

**Writing – original draft:** Zhong Chen.

**Writing – review & editing:** Wei Yang.

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
