## [Decision Letter · Decision Letter 0]

26 Dec 2023

PONE-D-23-32855Improved Assembly DC Circuit Breaker Based on Resonant Current InjectionPLOS ONE

Dear Dr. Cui,

Thank you for submitting your manuscript to PLOS ONE. After careful consideration, we feel that it has merit but does not fully meet PLOS ONE’s publication criteria as it currently stands. Therefore, we invite you to submit a revised version of the manuscript that addresses the points raised during the review process.

We look forward to receiving your revised manuscript.

Kind regards,

Chao Zhai

Academic Editor

PLOS ONE

Journal Requirements:

"The study was supported by “State Grid Corporation Headquarters Science and Technology Project (5500-20220110A-1-1-ZN)”"

"The study was supported by “State Grid Corporation Headquarters Science and Technology Project (5500-20220110A-1-1-ZN)"

"The study was supported by “State Grid Corporation Headquarters Science and Technology Project (5500-20220110A-1-1-ZN)"

5. We note that your Data Availability Statement is currently as follows: [All relevant data are within the manuscript and its Supporting Information files.]

Additional Editor Comments:

A major revision has to be made on the paper, and the authors should carefully address the issues raised by reviewers and improve the paper accordingly.

Reviewers' comments:

Reviewer's Responses to Questions

**Comments to the Author**

1. Is the manuscript technically sound, and do the data support the conclusions?

Reviewer #1: Yes

Reviewer #2: Yes

2. Has the statistical analysis been performed appropriately and rigorously? 

Reviewer #1: Yes

Reviewer #2: Yes

3. Have the authors made all data underlying the findings in their manuscript fully available?

Reviewer #1: Yes

Reviewer #2: Yes

4. Is the manuscript presented in an intelligible fashion and written in standard English?

Reviewer #1: Yes

Reviewer #2: Yes

5. Review Comments to the Author

Reviewer #1: 1. Provide specific parameter design specifications. This paper mentions CI-MPACB can cut off DC fault lines, but it does not provide specific details about the parameter design.

2. Resonant current injection is primarily designed for AC circuit breakers, DC circuits have a continuous and unidirectional flow of current, making it challenging to achieve the same resonant conditions. So how to achieve the same resonant conditions?

3. How Transient and Switching Stress are managed? In DC circuit breakers, resonant conditions may lead to high transient voltages and switching stresses. These transients can pose challenges in terms of insulation coordination and may require additional protective measures to prevent insulation breakdown and damage to the breaker components.

4. Achieving resonant current injection in DC circuit breakers often requires complex control algorithms and sophisticated electronic components. This increases the risk of malfunctions. So, what are the measures taken to rectify these malfunctions.

5. DC systems vary widely in terms of voltage levels, current ratings, and applications. Achieving a universal resonant current injection solution that is compatible with all DC systems is challenging, so how you will achieve this?

6. The additional complexity introduced by resonant current injection in DC circuit breakers may lead to increased maintenance requirements. So it may increase the cost. So, what is the remedy for this?

Reviewer #2: In this paper, a multiport assembly circuit breaker based on current injection (CI-MPACB) is proposed, which is able to generate a resonant current with increasing amplitude by controlling the duty cycle of Integrated Gate-Commutated Thyristors (IGCTs). Then the resonant current is injected into the SBS to generate current zero crossing and arc extinction. This article raises an interesting idea, but several issues must be addressed to clarify the work's novelty and contribution and some of the assumptions.

1. Existing simulations or results do not take into account the inductance of the DC line or the flat-wave reactor. Do they affect the simulation results?

2. There is no clear description of the working principle of the circuit breaker. It is recommended to combine time series and working process in detail.

3. The multi-port circuit breaker proposed in this paper has not been verified by simulation.

4. How to calculate the economy of multi-port circuit breaker?

5. Some experiments and results have not been compared with existing work. I think a comparison of results is needed to highlight the innovation and contribution of this paper.

6. It is also hoped that the author can show the current and voltage waveforms of T1 and T2 at work.

7. It is suggested to arrange the structure of the article reasonably in order to enhance the logic and readability of the article.

8. There are some mistakes in the units and variables in the article, please correct them in time.

6. PLOS authors have the option to publish the peer review history of their article (what does this mean?). If published, this will include your full peer review and any attached files.

Reviewer #1: No

Reviewer #2: No

---

## [Author Response · Author response to Decision Letter 0]

26 Feb 2024

Response to comments

We gratefully thank the editor and all reviewers for their time spent making their constructive remarks and useful suggestions, which has significantly raised the quality of the manuscript and has enabled us to improve the manuscript. Each suggested revision and comment, brought forward by the reviewers was accurately incorporated and considered. Below the comments of the reviewers are responses point by point and the revisions are indicated.

The supplementary and modified content of the article has been marked in red.

Reviewer #1:

1. Comment: Provide specific parameter design specifications. This paper mentions CI-MPACB can cut off DC fault lines, but it does not provide specific details about the parameter design.

1. Reply: Thanks for your valuable counsel. 

The content of designing specific parameters in sections 3.2 to 3.3 of the article includes the design of important resonant capacitor inductance, pre-charging capacitor and voltage parameters

2. Comment: Resonant current injection is primarily designed for AC circuit breakers, DC circuits have a continuous and unidirectional flow of current, making it challenging to achieve the same resonant conditions. So how to achieve the same resonant conditions?

2. Reply: Thank you for your comment.

This CI-MPACB can generate resonant current with an increased amplitude by controlling the duty cycle of the IGCT. When the current injection branch conducts, T2 is triggered in the first cycle. Cp continuously discharges outward. The current is injected into the branch to generate a clockwise flowing IC. And the current gradually increases. When the IC reaches its clockwise peak, the rate of change of the IC is zero. Due to the continuous current action of the inductor LH, the amplitude of the IC decreases in a clockwise direction. When the IC drops to zero, the rate of change of the IC is maximum and reaches its maximum value. At this point, the first peak is reached. The IC changes direction at zero point. When T1 is triggered, the IC will increase counterclockwise after zero crossing. When the IC reaches its counterclockwise peak, the polarity will change. The IC decreases from its counterclockwise peak until it reaches its peak. The specific principle analysis can be found on page 7.

3. Comment: How Transient and Switching Stress are managed? In DC circuit breakers, resonant conditions may lead to high transient voltages and switching stresses. These transients can pose challenges in terms of insulation coordination and may require additional protective measures to prevent insulation breakdown and damage to the breaker components.

3. Reply: Thank you for pointing this out. The image has been enlarged. 

Supplementary content：Both MB and AB use high-performance epoxy resin to seal the pole, which has good vacuum sealing, insulation performance, resistance to high and low temperature impacts, and mechanical impact performance.(page.4)

The MB adopts epoxy resin with excellent performance as the sealing pole to ensure the mechanical, electrical, and heat-resistant aging properties of the MB, ensuring good vacuum sealing performance, reliable insulation performance, resistance to high and low temperature impacts, and good mechanical impact performance.

4. Comment: Achieving resonant current injection in DC circuit breakers often requires complex control algorithms and sophisticated electronic components. This increases the risk of malfunctions. So, what are the measures taken to rectify these malfunctions.

4. Reply: Thanks for your valuable counsel.

Corresponding to the actual equipment, a forward zero crossing detection current transformer will be installed in the current injection branch. Control T1 and T2 by detecting the polarity of the resonant current. When there is an abnormality in the transfer branch submodule, an alarm signal is sent to the DC control and protection system. If the abnormal level exceeds the redundancy, the DC circuit breaker is not allowed to operate and an alarm signal is sent to the DC control and protection system.

5. Comment: DC systems vary widely in terms of voltage levels, current ratings, and applications. Achieving a universal resonant current injection solution that is compatible with all DC systems is challenging, so how you will achieve this?

5. Reply: Thank you for your comment

The MBS of the CI-MPACB can be used as a sub-module, which can be designed in series according to the voltage level of the system, so as to realize the application in different DC systems.

Supplementary content：(page.20).The CI-MPACB was applied to the ± 160 kV Nan ao multi terminal flexible DC transmission system. And compared with the MCB designed in paper [36]. Adopting a circuit breaker scheme with three 50kV sub-modules connected in series. The parameters of each sub-modules are selected as the optimization value mentioned above. The fault breaking waveforms are shown in the figure17.

Fig 17. Current waveforms of MB 

As shown in Fig. 17, the MB of the MCB proposed in the paper [36] starts to open and arc at the moment of 15 ms, and the current flowing through the MB continues to rise. The breaking current of the mechanical switch rises to 10.3kA at 18.5 ms. The commutator branch of the MCB is switched on. The precharge capacitor discharges and injects oscillating current into the MB. At the time of 18.89 ms, the current of MB oscillates to zero. The breaking time is 3.89ms. The peak value of superimposed current is 22.7kA. Under the same short-circuit condition, the MB of CI-MPACB starts to open at 16.5 ms. The current iniection branch is switched on at 18.5ms. The precharge capacitor discharges and T1 and T2 turn on alternately. At 18.8ms, the current of MB oscillates to zero and completes the breaking. The breaking time is 2.8ms. The peak superposition current is 20.07kA. The breaking time of CI-ACB is smaller than that of MCB. And the current stress is smaller.

Therefore, the CI-MPACB has a fast breaking speed and the injection current with an increasing periodic amplitude can form multiple zero crossing points, which can adapt to various short-circuit currents. This circuit breaker has high reliability in practical system applications.

And the CI-MPACB current injection branch designed in this paper can generate resonance current with rising amplitude to form multiple breakpoints, that is, it can break short circuit current under most different peak values.

6. Comment: The additional complexity introduced by resonant current injection in DC circuit breakers may lead to increased maintenance requirements. So it may increase the cost. So, what is the remedy for this?

6. Reply: Thank you for your significant reminding.

Compared with multiple branches corresponding to multiple CBs, the design of the new CI-MPACB proposed in this paper has greatly reduced the cost. The traditional ACB uses a large number of power electronic devices based on the hybrid circuit breaker. And in a multi-terminal system, because there is only one MBS with a power electronic device. The SBS contains only a MB. The advantages in the number of current injection branches greatly reduce the maintenance cost. So the new CI-MPACB proposed in this paper has a good economy.

Reviewer#2

1. Comment: Existing simulations or results do not take into account the inductance of the DC line or the flat-wave reactor. Do they affect the simulation results?

1. Reply: Thanks for your valuable counsel.

This article considers the inductance of DC transmission lines. The Bergeron model (page. 16) is mentioned in the paper. Without taking into account the flat wave reactor, the amplitude of the short circuit current and the rising speed will increase. The simulation results show that the circuit breaker designed in this paper can reliably break the high current scene.

2. Comment: There is no clear description of the working principle of the circuit breaker. It is recommended to combine time series and working process in detail.

2. Reply: Thanks for your valuable counsel. 

Sections 2.2 (page.4) and 2.3 (page.5) provide the specific working principle of the circuit breaker, include detailed time and corresponding action logic. Section 5.2 also shows the corresponding time and action logic of the circuit breaker simulation results (page.15).

3. Comment: The multi-port circuit breaker proposed in this paper has not been verified by simulation.

3. Reply: Thank you for your comment.

This article refers to a DC system with one sending end and multiple receiving ends to demonstrate the cost design advantages of this CI-MPACB. The simulation results of one end fault on the normal end transmission line have been reflected in this article (pages 18-19).

4. Comment: How to calculate the economy of multi-port circuit breaker?

4. Reply: Thank you for pointing this out.

The main focus of this article is to compare the cost of expensive power electronic devices. The price reference source has been added as [35] [Online] Available: https://octopart.com/. This CI-MPACB has significant advantages over multi hybrid DCCB, as well as traditional power electronic device circuit breakers based on hybrid DC circuit breakers.

5. Comment: Some experiments and results have not been compared with existing work. I think a comparison of results is needed to highlight the innovation and contribution of this paper.

5. Reply: Thank you for your comments on our article.

Supplementary content：(page.20)：The CI-MPACB was applied to the ± 160 kV Nan ao multi terminal flexible DC transmission system, and compared with the MCB designed in paper [36]. Adopting a circuit breaker scheme with three 50kV sub-modules connected in series. The parameters of each sub-modules are selected as the optimization value mentioned above. The fault breaking waveforms are shown in the figure17.

Fig 17. Current waveforms of MB 

As shown in Fig. 17, the MB of the MCB proposed in the paper [36] starts to open and arc at the moment of 15 ms, and the current flowing through the MB continues to rise. The breaking current of the mechanical switch rises to 10.3kA at 18.5 ms. The commutator branch of the MCB is switched on. The precharge capacitor discharges and injects oscillating current into the MB. At the time of 18.89 ms, the current of MB oscillates to zero. The breaking time is 3.89ms. The peak value of superimposed current is 22.7kA. Under the same short-circuit condition, the MB of CI-MPACB starts to open at 16.5 ms. The current iniection branch is switched on at 18.5ms. The precharge capacitor discharges and T1 and T2 turn on alternately. At 18.8ms, the current of MB oscillates to zero and completes the breaking. The breaking time is 2.8ms. The peak superposition current is 20.07kA. The breaking time of CI-ACB is smaller than that of MCB. And the current stress is smaller.

Therefore, the CI-MPACB has a fast breaking speed and the injection current with an increasing periodic amplitude can form multiple zero crossing points, which can adapt to various short-circuit currents. This circuit breaker has high reliability in practical system applications

6. Comment: It is also hoped that the author can show the current and voltage waveforms of T1 and T2 at work.

6. Reply: Thank you for your significant reminding.

Supplementary content：T1 and T2 electrical quantity waveforms (page.15)。

(a) waveforms of current

(b) waveforms of voltage

Fig 12. T1 and T2 electrical quantity waveforms

As shown in Figure 12, the current superposition of T1 and T2 tubes is the current of the current injection branch. And the voltage amplitude of T1 and T2 tubes decreases with the increase of turn-on times.

7. Comment: It is suggested to arrange the structure of the article reasonably in order to enhance the logic and readability of the article.

7. Reply: Thanks for your valuable counsel.

The content has been added according to the comments and placed in the appropriate position of the article.

8.Comment: There are some mistakes in the units and variables in the article, please correct them in time.

8. Reply: Thank you for pointing this out.

---

## [Decision Letter · Decision Letter 1]

22 May 2024

Improved Assembly DC Circuit Breaker Based on Resonant Current Injection

PONE-D-23-32855R1

Dear Dr. Cui,

We’re pleased to inform you that your manuscript has been judged scientifically suitable for publication and will be formally accepted for publication once it meets all outstanding technical requirements.

Kind regards,

Arvind R. Singh, Ph.D

Academic Editor

PLOS ONE

Additional Editor Comments (optional):

Accept

Reviewers' comments:

Reviewer's Responses to Questions

**Comments to the Author**

1. If the authors have adequately addressed your comments raised in a previous round of review and you feel that this manuscript is now acceptable for publication, you may indicate that here to bypass the “Comments to the Author” section, enter your conflict of interest statement in the “Confidential to Editor” section, and submit your "Accept" recommendation.

Reviewer #1: All comments have been addressed

Reviewer #2: (No Response)

2. Is the manuscript technically sound, and do the data support the conclusions?

Reviewer #1: Yes

Reviewer #2: (No Response)

3. Has the statistical analysis been performed appropriately and rigorously? 

Reviewer #1: Yes

Reviewer #2: (No Response)

4. Have the authors made all data underlying the findings in their manuscript fully available?

Reviewer #1: Yes

Reviewer #2: (No Response)

5. Is the manuscript presented in an intelligible fashion and written in standard English?

Reviewer #1: Yes

Reviewer #2: (No Response)

6. Review Comments to the Author

Reviewer #1: (No Response)

Reviewer #2: (No Response)

7. PLOS authors have the option to publish the peer review history of their article (what does this mean?). If published, this will include your full peer review and any attached files.

Reviewer #1: No

Reviewer #2: No

---

## [Editor Report · Acceptance letter]

7 Jun 2024

PONE-D-23-32855R1 

PLOS ONE

Dear Dr. Cui, 

I'm pleased to inform you that your manuscript has been deemed suitable for publication in PLOS ONE. Congratulations! Your manuscript is now being handed over to our production team.

Kind regards, 

on behalf of

Dr. Arvind R. Singh 

Academic Editor

PLOS ONE